# A New Model of Learning: Environmental Health in a Global World

**DOI:** 10.3390/ijerph20126146

**Published:** 2023-06-16

**Authors:** William N. Rom, Aishwarya Rao, Lori Hoepner, Chris Dickey

**Affiliations:** 1Department of Global and Environmental Health, NYU School of Global Public Health, 708 Broadway, New York, NY 10003, USA; ar6666@nyu.edu (A.R.); chris.dickey@nyu.edu (C.D.); 2School of Public Health, SUNY Downstate Health Sciences University, Brooklyn, NY 11203, USA; lori.hoepner@downstate.edu

**Keywords:** Environmental Health, global, student learning

## Abstract

Introduction. Environmental Health in a Global World at New York University was re-designed as a class participatory effort, challenging undergraduate students to understand environmental hazards and the resultant adverse health outcomes by embracing the inherent complexity of environmental risks and proposing solutions. Methods. Following introductory lectures, students are placed into teams and assigned a specific perspective, or avatar, which includes learning to see the challenge from the perspective of a technical expert such as a biologist, an engineer, or an anthropologist. The teams then design specific systems maps to visualize the complex interactions that lead to adverse health outcomes after a given environmental exposure. The maps highlight potential leverage points where relatively minor interventions can provide a disproportionate benefit in health outcomes. The teams then explore potential interventions and identify the potential unintended consequences of those actions, develop and advocate for innovative new strategies to mitigate risk and improve outcomes. Results and Discussion. Over the past 5 years, we have taught this methodology to over 680 students with strong, student-oriented results. The teams created and presented more than 100 strategies, addressing a diverse set of environmental challenges that include water contamination, gun violence, air pollution, environmental justice, health security, and climate change. Developing the strategies helped the students understand environmental threats in a more holistic way, provided them with some agency in finding solutions, and offered an opportunity for them to improve their presentation skills. The responses in course evaluations have been enthusiastic, with many students reporting a deep impact on their college experience.

## 1. Introduction

In the academic years 2017–2018, we re-designed the undergraduate environmental health course that is required of all public health co-majors at New York University. Our idea was to develop a course that develops skills that are needed to address a wide spectrum of pressing environmental health challenges, along with opportunities for students to propose their own strategies to mitigate the health effects of environmental factors. At the outset, we informed the students that we were trying a new way of teaching environmental health and that their feedback was important to us. This created an unusual partnership between the faculty and the students in trying to deliver an effective course. The result is a novel learning structure that can be replicated across multiple public health disciplines.

## 2. Methods

The class starts with an introduction to systems thinking and the visual representation of the factors that contribute to complexity in systems [1]. We base it on the premise that all environmental health challenges are complex and the result of multiple political, social, economic, and other factors. By attempting to understand the interactions between these factors, students can evaluate multiple strategies to improve health outcomes and analyze both the intended and unintended consequences of their proposed actions. Through a series of modules, teams create a map of the system that produces an adverse health outcome (e.g., increased blood lead levels in children) resulting from environmental exposure (e.g., lead in water). The team then develops a strategy to improve that outcome based on the interactions among the factors highlighted in their systems map. They then present that strategy to the rest of the class for peer review and feedback. Through this methodology, the students develop skills in critical systems thinking and presentation while developing a deep understanding of the environmental challenges at hand. The modules include smoking and lung cancer, lead in water (using Flint, MI as a case study), exposure to air pollution, gun violence and injuries, climate change and food security, health security (based on World Health Organization (WHO) Joint External Evaluation reports (JEE)), and planetary health. Teams are asked to confine their analysis to a geographical region and at-risk population.

To add another layer of complexity, and to help ensure that the teams consider multiple perspectives, each member of a team is assigned a role with a particular viewpoint on the challenge. Team members are randomly assigned to approach the problem from the perspectives of anthropologists, economists, policy/government officials, epidemiologists, biologists or engineers, and environmental scientists. We call these roles the students’ avatars.

Our initial concerns included the following:Would students embrace the concept of systems thinking and learn to draw a systems map that shows the interaction between the complex factors that influence the outcome of a system?Would they engage in the level of cooperation and research necessary to understand an environmental topic, lead in water, for example, enough to draw a useful systems map?Would they be comfortable analyzing their map to propose potential strategies for improving outcomes?Could they learn enough about their avatar to feel confident in expressing that perspective?Could they effectively create a persuasive case for implementing an innovative strategy to reduce adverse health outcomes related to an environmental challenge?Would they be able to accomplish that seven times in a semester and maintain a high standard of performance?

## 3. Results

We taught the revamped course beginning in the fall of 2018 to a wide spectrum of students (total *n* = 680) in two classes. The students were mostly upper classes: 30% senior, 40% junior, 28% sophomore, and 2% freshmen. We have since taught the course to more than 180 students each year. Interestingly, practically all had dual majors in sociology, anthropology, nutrition, nursing, biology, social work, media, and history.

Smoking and lung cancer was the first module for the class to participate in groups with various roles to play: anthropologist, epidemiologist, engineer, environmental scientist, economist, and policy/government official. These roles provide an opportunity for the student to envision the complexities and interrelationships to environmental problems. For this first assignment, they are challenged to each produce a systems map of smoking and how it is to be proscribed in society and public health. Groups of five to six students are assigned specific module topics for their group to research and present a slide talk (1 slide per student) to the class. The six topics were: (1) lung cancer and smoking; (2) cardiovascular disease and smoking; (3) chronic obstructive pulmonary disease and smoking; (4) second-hand smoke; (5) methods of quitting smoking; (6) e-cigarettes positive trends; and (7) e-cigarettes negative trends. Figure 1 illustrates a smoking and lung cancer systems map.

The second module was lead in drinking water. The lead/water groups were: Flint Lead Crisis; Nitrates-Phosphates; PFAS Per-and poly-alkane substances; Cryptosporidioisis-Vibrios; Fracking; Waters of the United States EPA rule; Legionnnaires Disease. The third module was air pollution, and the groups were: Particulate matter < 2.5. Five microns; Beijing air pollution; traffic-related air pollution; ozone; New Delhi air pollution; sulfur dioxide and acid rain; and nitrogen oxides, especially gas stoves. Fourth, the groups for climate change were China; India; the United Kingdom; Germany; South Africa; Brazil; and Canada, covering the major emitters of greenhouse gases. Fifth, the gun module was divided into seven topics: National Rifle Association; gun control (Brady Campaign); background checks; mental health; research by government agencies, e.g., the CDC and NIH; gun technologies; and New York State and City gun laws and regulations. Sixth, the health security class projects were: Ebola (Democratic Republic of the Congo), Zika (Brazil), COVID-19 (China), polio (Pakistan), HIV (South Africa), cholera (Yemen), and Bhopal methyl isocyanate exposure (India). The seventh module was planetary health with several class projects: Arctic National Wildlife Refuge, Bears Ears National Monument, protecting Sage Grouse, Boundary Waters Canoe Area Wilderness protections, Serengeti National Park, Amazon Rainforest, and New York’s Adirondack State Park. The teams are assigned at random for the first two modules, then re-assigned for the third and fourth modules, and finally re-assigned for the last three modules. In special cases, we take into consideration special requests for specific assignments. There are 4 points each for the presentation, memo, and systems maps for the 7 modules, plus 16 points for the final reflection memo.

Memo 1 illustrates a group memo.

**Memo 1:** *Example of a Student Memo*.

### Traffic-Related Air Pollution Memo

Background (Environmental Scientist).

Traffic-related air pollution (TRAP) is often the cause of many adverse health effects related to respiratory and cardiovascular issues in urban areas. One of the major sources of air pollution is traffic due to the numerous pollutants that can be released from vehicles, such as “evaporative emissions from vehicles, and non-combustion emissions (e.g., road dust, tire wear)”, as well as vehicle exhausts. Exposure to TRAP is often associated with numerous adverse health effects, such as asthma attacks in young children; impaired lung function; a potential increase in cancer risk; an increased prevalence of hypertension; premature death due to cardiovascular disease; and negative impacts on fetuses during pregnancy leading to low-birthweight, childhood leukemia, and an increase in neonatal infant mortality. TRAP proves to be very problematic as it is documented that “the number of people living ‘next to a busy road’ may include 30 to 45 percent of the urban population in North America”. Children and teenagers are the most vulnerable but not the only ones at risk of exposure to TRAP, as drivers, pedestrians, bicyclists, railway workers, and individuals who live next to busy roads are also subject to air pollutants. It is quite difficult to determine whether some adverse health effects are caused by TRAP or other pollution sources, but there is research that continues between exposure to TRAP and negative health outcomes.

Epidemiologist.

While TRAP affects a multitude of populations, there have been many studies conducted on the association between TRAP and childhood respiratory illness. It was found that for children residing at their current address in San Francisco for at least 1 year, there was found to be modest but significant increases in the odds of bronchitis symptoms and physician-diagnosed asthma in neighborhoods with higher concentrations of traffic pollutants. School positioning was also associated with respiratory illness. Concentrations of several pollutants (Black Carbon, NOX, NO, and, to a lesser extent, NO_2_) were higher at schools located within 300 m downwind of a freeway compared with those at schools upwind or further from major traffic sources. Additionally, TRAP can be associated with adverse effects on pregnancy. It is suggested that higher traffic-related air pollution levels are associated with pregnancy loss, with the strongest estimates between the 10th and 20th gestational weeks. In Boston, the strongest association was during the 15th week of gestation; for every 10 ppb of NO_2_ increase during that week, there is an observed lower rate of live births.

Anthropologist.

Communities of color and lower income are disproportionately affected by traffic-related air pollution due to historic housing practices and lack of access to healthcare resources. Morris Heights in the West Bronx is the community with the highest level of traffic-related air pollution in New York State. The population of this community is almost entirely persons of color—70% Latino, 29% African American. Although this population is most susceptible to vehicle-pollution-related illness due to the location in which they live, they are less likely to have access to adequate healthcare services to treat illnesses from air pollution. While higher-earning professionals who can afford to live outside the city commute to work daily and pass through neighborhoods such as Morris Heights, residents of neighborhoods located near highways are left to face the negative effects of traffic congestion.

Policy/Government Official.

The increasing implementation of congestion pricing plays a major role in the reduction in traffic-related air pollution. The U.S. Department of Transportation defines congestion pricing as “a way of harnessing the power of the market to reduce the waste associated with traffic congestion”. The approach can work in various ways, including changes in pricing regarding tolls in a specific lane, region, or more. Congestion pricing aims to encourage drivers to travel during off-peak times and in less dense areas to reduce pollution and bolster public transportation. Currently, congestion pricing has been adopted in Stockholm and London, with New York City slated to join the movement. In fact, in March of 2021, New York State received approval from the Biden Administration U.S. Department of Transportation to proceed with the “Environmental Assessment and public outreach” required to implement the congestion pricing program. Effective policy would include support for these congestion pricing programs to improve public transport and population health and decrease overall air pollution.

Economist.

The economic costs of traffic-related air pollution can be assessed in multiple ways. First, the illnesses that come as a direct consequence of pollution have a large financial burden associated with them. Studies conducted in Warsaw, Poland, found that approximately “827 Warsaw citizens die in a year as a result of traffic-related air pollution [and] about 566 and 250 hospital admissions due to cardiovascular and respiratory diseases, respectively, and more than 128,453 restricted activity days can be attributed to the traffic emissions”. Altogether, the costs of the illnesses total about USD 403 million. Further, the cleanup and alternative solutions may also be costly. The highly sought high-speed rail system, which has been implemented in China, would cost the United States about USD 4 trillion to replicate. In California alone, a 520-mile route from San Francisco to Los Angeles is projected to cost USD 100 billion. In the meantime, the street sweeping system on highways that helps remove extra debris and dust left by traffic is also rather expensive.

Engineer.

To reduce traffic-related air pollution, one can simply choose biking or walking over driving their cars to prevent congestion of vehicles on the road. However, this option is not readily available to many people for a variety of reasons. Still, there are several engineering solutions that have been implemented in the past 10 years and show promise of reducing traffic-related air pollution. One of these solutions is the use of alternate fuels. Most pollution-creating cars use petroleum fuel. The best alternative to this is hydrogen fuel cells. This technology works by using hydrogen gas to create electricity. This is then converted to mechanical energy in an electric motor to get the engine moving and, thus, the car running. Advantageously, the only emission of hydrogen cell fuels is water. Another engineering solution to reduce air pollution would be the use of electric cars. Within the past decade, electric cars have become more and more popular, but surprisingly, electric cars were introduced in the 1890s.

## 4. Discussion

We began the course with an Introduction to the Environment: Silent Spring by Rachel Carson, published in 1962, highlighting the indiscriminate use of pesticides [2]. Carson accused the chemical industry of spreading misinformation and public officials of accepting the industry’s marketing claims unquestionably. She studied DDT and found that it led to the death of countless other species, especially birds, in its application to agricultural fields. She also worked with Wilhelm Hueper at the National Cancer Institute on the pesticide-cancer connection. She was prescient about the environmental effects, even though many effects were described later, e.g., the effects of DDT on egg thinning and the reproductive collapse of the bald eagle. Oil spills from off-shore platforms fouling the beaches of Santa Barbara, California, and fires raging atop the oil-slicked Cuyahoga River as it traversed the City of Cleveland gave rise to a vigorous bipartisan environmental movement captured by Senator Gaylord Nelson, Founder of Earth Day 1970. Over 20 million Americans participated in this auspicious event [3]. Additionally, in the beginning, we demonstrate how to use the metrics of the Global Burden of Disease and its relationship to environmental exposures [4].

An introduction to environmental epidemiology orients the students to study diseases and health conditions occurring in the population that are linked to environmental factors. We cover definitions of prevalence, incidence, and case fatality rate. Important historical observations are covered, e.g., John Snow’s removal of the handle on the Broad Street pump where most of the cholera victims had obtained their drinking water, and Percival Pott’s finding the cancer of the skin of the scrotum was commonplace in chimney sweeps who cleaned the coal dust-laden London chimneys. Case reports, case series, ecologic studies, odds ratios from case-control studies, and relative risk from cohort studies are reviewed. In this regard, we use Sir Austin Bradford Hill’s President’s Address to the Section of Occupational Medicine for Association or Causation: strength of the association; consistency; specificity; temporality; biological gradient; plausibility; coherence; and experiment [5]. Three papers are used to illustrate Environmental Epidemiology: Richard Doll’s classic study of 113 men working in Turner Brothers’ Asbestos Works >20 years with 11 lung cancers observed and 0.8 expected [6]. The calculation of expected rates was a challenge. Dr. Irving J. Selikoff faced similar challenges in New York, where he gathered 632 asbestos workers from the Union Locals 12 and 32 records and identified those with >20 years’ exposure observing 45 lung cancers with 6.6 expected; expected rates were from E. Cuyler Hammond of the American Cancer Society where they enrolled study subjects in a prospective study in the 1950s [7]. Since smoking was a confounding factor introducing bias to the students, we used Dr. Selikoff’s follow-up study to evaluate this. Over the next four and a half years, he followed the surviving asbestos workers (*n* = 370) to interrogate smoking habits and calculate expected rates as if they had not smoked or worked with asbestos. He found a more than additive risk if study subjects had both smoked and worked with asbestos increasing their lung cancer risk by 92 times [8].

*The New England Journal of Medicine* publishes Clinical Problem-Solving case reports that we utilize for toxicology: a sickening tale of arsenic exposure and poisoning [9].

Protection from environmentally associated health hazards is taught as a fundamental human right. The year 1970 was a key year in the development of environmental agencies, laws, and regulations. The Executive Branch formed the Environmental Protection Agency, which could execute the laws about clean air, water, and the environment. EPA was founded in December of that year with William Ruckelshaus as its first Administrator. He brought personnel and agencies together to address radiation, clean air regulations, pesticides, water pollution and sewage treatment, and protection of the Great Lakes and oceans. The National Environmental Policy Act (NEPA) was enacted in 1970, mandating the federal government to conduct an Environmental Assessment or Environmental Impact Statement stating how a project will be built, the consequences on local communities, alternative ways to develop the project that fulfills the needs of the government but causes less harm to people or the environment, and measures that can be taken to lessen any harmful impacts of the project. The Clean Air Act, the bedrock environmental law, was established fundamentally as a public health law. The Clean Air Act called for a 90% reduction in automobile emissions by 1975 and the best available control technology on power plants and industrial sources. This was a challenge, resulting in the 1990 Clean Air Act Amendments: they established National Ambient Air Quality Standards to protect health and, secondly, to protect welfare interpreted as the environment. There were six standards: particulate matter (eventually focused on particles <2.5 microns), carbon monoxide, lead, sulfur dioxide, ozone, and nitrogen oxides. At the 50-year anniversary of the EPA, they registered a 30:1 ratio of benefits to costs since 1990; USD 85 billion in costs and USD 2 trillion in benefits. In 2020, we averted 230,000 deaths, 200,000 heart attacks, 2,400,000 asthma attacks, 120,000 ER visits, and 17,000,000 lost workdays. Air pollutant measurements of six criteria pollutants consistently declined as the Gross Domestic Product increased. The Clean Water Act (1972) charged the EPA to develop Water Quality Standards for 126 priority pollutants and define waters under its jurisdiction by their chemical, biological, and physical nexus.

There were seven student modules illustrating environmental exposures and diseases. First, smoking continues to be a public health hazard even though active smoking is declining in the U.S. (only 11% active smokers), and ~440,000 U.S. deaths are attributable to this habit. Globally there are a billion and a half consumers of tobacco products who smoke over 6 trillion cigarettes and dispose of 1.7 billion pounds of cigarette butts. More than 40% of smokers reside in China and India. Half of smokers can expect to die prematurely, and even for those who smoke 10 or fewer cigarettes per day, life expectancy is shorter, and lung cancer risk is 20 times higher than never-smokers [10]. Second-hand smoke increases the risk of contracting lung cancer by 30% and coronary heart disease by 25%. The Surgeon General 1964 Report concluded smoking was a cause of lung cancer in men and chronic bronchitis; ever since this report, smoking prevalence declined. The 1972 Surgeon General Report called attention to second-hand smoke causing respiratory illness and symptoms in children of smokers, and in 1986 concluded second-hand smoke was a cause of lung cancer in nonsmokers [11]. There are over 60 carcinogens in cigarette smoke, with major adducts and mutations occurring from acrolein and acetaldehyde, N-nitrosamines, and polycyclic aromatic hydrocarbons, including benzo(a)pyrene [12].

Second, lead in drinking water in Flint, MI, is used as a case study as an abject failure to protect public health. Flint had a financial crisis in 2014 following the departure of 80,000 automotive jobs leaving 5000 remaining and an impoverished minority community behind. The governor appointed an emergency manager who decided to save money by switching the public water supply from Detroit’s source, Lake Huron, to the local Flint River. An alert pediatrician, Dr. Mona Hanna-Attisha, measured blood lead in her patients, finding an increase from 4% to 10.6% above the reference value of 5 micrograms per deciliter in those homes with the highest lead levels in the water [13]. In 2017 EPA announced that it awarded USD 100 million to Flint for drinking water infrastructure upgrades. By the end of 2020, more than 26,000 excavations were completed, resulting in the identification and replacement of over 9700 lead service lines. The classic *New England Journal* study by Herbert Needleman is reviewed; he collected baby teeth from over >3200 first and second-graders in Chelsea and Somerville, Massachusetts [14]. The parents brought the schoolchildren to a testing center where neurobehavioral studies were conducted, and the teachers blindly scored the attention and behavior of their students. Lead was measured in the teeth, and the 58 highest were compared to the 100 with the lowest dentine levels. Children with high lead levels scored significantly less well on the Wechsler Intelligence Scale for Children, lower on verbal subsets, on three other measures of auditory and or speech processing, on a measure of attention, and IQ. The Flint, Michigan, water was also a source of Legionnaires’ disease, with a statistical relationship between low chlorine residual in the drinking water system and the outbreak [15]. Kenneth Bridbord and David Hanson wrote an elegant treatise on how they were able to develop an EPA regulation that resulted in the removal of tetra-ethyl lead from gasoline. Their success was paralleled by a significant drop in the nation’s children’s blood lead levels from over 15 to less than 10 micrograms per deciliter in the 1980s and 1990s [16].

The third module was air pollution, with a focus on particulate matter <2.5 microns. Air pollution comes from fossil fuel burning in coal and natural gas electricity-generating power plants, industry, and in transportation. Globally, over 3.5 million people die from outdoor air pollution, and a similar number die from indoor air pollution. Particulate matter under 2.5 microns that is inhaled deep into the lungs is associated with increased cardiopulmonary mortality and lung cancer. Studies are reviewed in the class that build the scientific literature against PM_2.5_ and ozone: the American Cancer Society cohort assembled in the 1970s and 1980s, the Harvard Six Cities Studies that enrolled over 8000 residents and compared their mortality to PM_2.5_ measurements, the time series studies of the National Mortality and Morbidity Study [17,18,19]. There is a straight line in the hazard ratio for cardiopulmonary mortality and PM_2.5_ exposure with our current standard of 12 micrograms/m^3^ [20]. The standard should be low enough to protect vulnerable people in society, e.g., the elderly, those with chronic illness, and children. Sixty asthmatics who walked either on a polluted street in London versus a walk in the park had reduced lung function and more inflammatory mediators in sputum after the walk along diesel-polluted Oxford Street [21]. Particulate matter instilled into the lungs of normal volunteers had bronchoalveolar lavage with increased inflammatory mediators [22]. Transgenic mice exposed to PM_2.5_ chronically developed accelerated atherosclerosis [23].

The fourth module was climate change and global public health. The greatest threat to global public health is global warming and the continued failure of world leaders in government and business to keep the global temperature rise below 1.5 °C and to restore nature [24]. Global warming follows the reflection of infrared radiation back to the surface by a greenhouse gas layer above the Earth. We have known that CO_2_ and greenhouse gases could warm the planet for more than a century since Svante Arrhenius postulated the greenhouse gas effect in 1898. The World Meteorological Association has kept global temperature readings since 1888, and we now are 1.1 °C above the global mean. The heat has been greater in the Northern Hemisphere, with the Arctic warming at >2 °C. There has been a concomitant increase in CO_2_: from 312 ppm in 1959 to 420 ppm today. The increase has been measured by David Keeling on top of Mauna Loa in Hawaii called the Keeling Curve. There are other important greenhouse gases, including methane (CH_4_). Methane is from oil and gas wells, landfills, and animal releases. It is 80 times more potent than CO_2_ over a time span of about 20 years. CO_2_ is a long-lived gas with a residence time of 33% remaining after 100 years and 20% at 1000 years. Hydrochlorofluorocarbons and hydrofluorocarbons used in coolants are global heating gases with over 2000 times the potency of CO_2_. Ice core data from Antarctica with measurements of CO_2_ in bubbles and deuterium as a proxy for temperature show that there have never been CO_2_ levels above 300 ppm for 800,000 years. The source of global warming is from fossil fuels, where we release over 36 billion metric tons of CO_2_ per year. Since 1970, there has been a gigantic increase in coal, oil, and natural gas that has heated the planet, referred to as Anthropogene. The major emitters are China 32%, the United States 14%, Europe 8%, and India 8%. The CO_2_-emitting sectors are transportation, electricity, industry, buildings, and agriculture. The ocean absorbs the most heat; the additional heat is equivalent to 5 Hiroshima atomic bombs exploding per second for 25 years. Heat waves and increased days of extreme heat are the first health consequence of global heating [25]. In 2020 there was a heat dome over the Pacific Northwest, and over 1000 people died; in addition, extreme drought-exacerbated wildfires have even consumed entire small communities. The exemplary heat wave occurred in France in August of 2003. The mean maximum temperature exceeded the seasonal norm by 11–12 °C on nine consecutive days with little cooling at night. There were 15,000 excess deaths observed in France (a 60% increase over expected mortality) and 32,000 in Western Europe, with >70,000 extended in Europe. There were increases in heatstroke, hyperthermia, and dehydration; increases in heart failure, chronic respiratory disease, and stroke also occurred, especially among the elderly and those living alone. A lack of air conditioning was critical. Extreme weather is a public health threat since global warming increases evaporation, and the increased clouds may result in heavier storms. Droughts severely limit agriculture, and in extreme situations, e.g., in southern Madagascar, result in famine. Further, drought in Guatemala, Honduras, and El Salvador is a leading cause of climate migration north to the U.S. border. Hurricane Sandy cost over USD 60 billion in property losses in New York and New Jersey and over 60 deaths. In Puerto Rico, there were over 4000 deaths during and following Hurricane Maria. Many of these deaths were due to the inability to reach medical care and refill prescriptions, for example.

Glacial ice melting will drastically alter the environs of the polar regions and contribute to sea level rise. There are tipping points such as the loss of coral reefs, melting of Greenland and Antarctic ice sheets (they contain 60 m of sea level rise), thawing of the permafrost layer, or loss of the Amazon rainforest due to logging and clearing of land for cattle ranching and soybean crops (if 20% of the Amazon rainforest were lost, the forest would become a CO_2_ source rather than a carbon sink). Solving the climate crisis is a commitment to staying below 1.5 °C; the goal is to reach 45–50% CO_2_ levels below the 2005 level by 2030 and net zero by 2050.

## 5. Climate Change Presentation Slides in Online Appendix A

We provide a session on the ozone hole as an example of how the international community can work together and with industry to solve an unexpected environmental problem of the first order [26].

The fifth module focuses on gun violence and injuries. Gun violence is a serious public health problem, with 120,000 individuals shot in America every year and 45,000 killed (Kristoff [27]). It is an emergency room problem and surgery challenge with bullets exploding to cause serious organ damage and long-lasting physical and mental disabilities. It is a justice problem, and it affects young people and their future families. More Americans have died from gun violence (1.4 million) since 1975 than in all the wars in American history going back to the Revolutionary War. More preschoolers are shot dead in America (about 75) than police officers are in a typical year. Across the nation, there are 200–500 justifiable homicides involving a private citizen using a firearm reported to the FBI (1.6%); that same year, there were 10,341 criminal gun homicides. That owning a gun to protect oneself against criminals is a myth. A gun in a household increases the risk of a gun death. The USA has more guns than any other country: 88.8 per 100 people compared to 45.7 in Switzerland, 30.8 in Canada, and 0.6 in Japan. We have a gun death every 15 min. Deaths in mass shootings are only 1.2% of gun deaths in the U.S. A public health approach to guns should include the following with a goal to reduce firearms deaths in America by one-third—10,000 or more lives saved each year: background checks (22% of guns are obtained without one); protection orders (keep men subject to domestic violence protection orders from having a gun); ban those under age 21 from buying a gun; safe storage in locked cases and store guns/ammunition separately and locked; prevent straw purchases; ammunition checks (one-time background check); end immunity for firearm companies; ban bump stocks; research smart guns such as a fingerprint or PIN entered; gun buy-back program for local police departments; ban military-style automatic or semi-automatic AR-15 rifles [27]. States with weak gun laws or more guns per household have much greater death rates from gun violence; counties with increased prevalence of firearms have greater violent crime, homicide, rape, and robbery. A public health approach is necessary to incorporate principles of responsible and safe firearm ownership into the legal interpretation of the 2nd amendment to ensure a safer future.

The sixth module was health security. We began with an overview of international health regulations and the response by the United States and several other developing countries to meet those standards [28]. This was followed by WHO recommendations on vaccines, an introduction to vaccine hesitancy, and emerging epidemics of Ebola and Zika. These beginnings were completely overwhelmed by the COVID-19 pandemic resulting in expanded material on this topic. International Health Regulations were designed to protect against the international spread of public health emergencies with unnecessary interference with global travel and trade [28]. The regulations require member states to develop national health surveillance and response systems. The revised regulations required member states to assess their core capacity for effective public health response and surveillance to assess any event within 48 h. WHO is the primary arbiter on decisions relating to controlling public health threats. In the U.S., The Pandemic and All-Hazards Preparedness Act created an Assistant Secretary of HHS to advise on federal public health medical surveillance and response, developed a National Health Security Strategy, and provided financial assistance to poorer countries to develop public health capacity. It created the Biomedical Advanced Research and Development Authority (BARDA) within HHS to foster the rapid development of drugs and vaccines against highly infectious pathogens. We cover childhood vaccines: DPT, MMR, polio, and others (Varicella-Zoster, hepatitis B, papillomavirus). In the U.S., for a single birth cohort, vaccines prevented 20 million cases of disease and 40,000 deaths. Vaccines were a primary driver of global health, with 118 million infants vaccinated in 2018 driving a 55% decline in <5 year mortality. WHO considered vaccine hesitancy a top 10 global health threat. The global health security agenda was driven by the emerging threats posed by HIV, SARS, H1N1 flu, Ebola, and Zika. Now there are 37 million people living with HIV/AIDS, and concomitant epidemics of tuberculosis and MDR-TB in New York City in the 1990s and South Africa. The President’s Emergency Plan for AIDS Relief (PEPFAR) has saved 11 million lives over 15 years in Africa, and the Global Fund to Fight AIDS, Tuberculosis, and Malaria provides almost USD 1.35 billion/ year of health assistance.

From December 2019 to January 2020, we had a pandemic with severe acute respiratory syndrome coronavirus 2 (SARS-CoV-2) causing COVID-19 [29]. The outbreak began in Wuhan, China, and was probably a jump of a mutated coronavirus from wild animals spread to humans in a live wild animal food market (although alternative origins from a laboratory leak from a coronavirus-infected worker). Individuals exposed in the market developed severe pneumonia with high fever, fatigue, cough, and breathlessness. It spread to others before it was recognized as a new virus and caused numerous severe respiratory illnesses. CT scans showed it to be extremely inflammatory and bilateral, with the pneumonia rapidly advancing to respiratory failure. Chinese authorities tried to minimize the threat, but emissaries from China’s CDC recognized it to be a novel virus and that it spread person-to-person. The world reacted by quarantines and massive lockdowns since there was no effective treatment. By 10 January 2020, the Chinese had sequenced the viral RNA and showed that it was an enveloped RNA virus that had a large genome of 30,000 kb. By May 2023, the WHO reported 766 million confirmed cases and 6,932,591 deaths with over 13 billion vaccine doses administered; in the U.S., about a third of the population developed cases with 1.1 million deaths despite 70% receiving two or more vaccine doses. Soon it was recognized that transmission was by respiratory aerosols and that N95 masks were highly preventive against transmission. Restaurants, gyms, and bars, as well as in-person meetings, hastened the spread and even caused super-spreader events. Of those with symptoms, 81% were mild–moderate, with almost half being asymptomatic, but severe symptoms were seen in 15%, and 5% required ICU care. There was an initial case–fatality rate of 1.6–2.3% (much worse than flu with a case–fatality rate of 0.1%). Obesity, diabetes mellitus, COPD, heart disease, and kidney disease increased risk, especially for those who were immuno-compromised. The disease syndrome included neurological disorders, hyperinflammation, cardiac dysfunction, hypercoagulability and pulmonary emboli, and multisystem inflammatory syndrome in children. Case and hospitalization rates were higher among Hispanics, American Indians, and Blacks. Monoclonal antibodies, the anti-viral Paxlovid, oxygen, and dexamethasone were mainstay treatments. The coronavirus used a receptor binding domain at the end of its spike protein to bind human cells using the ACE Receptor (angiotensin-converting enzyme-2), which was a key site for neutralizing antibodies to target the virus. Mutations in the virus’ spike region have created variants that are much more transmissible. Vaccines using RNA technology targeting the spike protein were over 95% effective at preventing COVID-19 infection, hospitalization, and death in clinical trials; two companies (Pfizer (New York, NY, USA), Moderna (Cambridge, MA, USA)) were funded by the U.S. government to produce millions of doses. Lagging were efforts at vaccination in developing countries, especially in Africa. Global efforts were made to deliver vaccines to these countries, but challenges remained since RNA vaccines must be kept at extremely cold temperatures, and the two-dose regimen required many public health workers. Stubbornly, a third of individuals in developed countries were laggards at getting vaccines, and misinformation flourished on social and right-wing media challenging public health authorities on vaccines and masks.

The seventh module was planetary health, where we focused on public land policies leading to the protection of nature for biodiversity and human health/welfare [30]. The goal was to protect 30% of land and water resources by 2030 (30 × 30). Humans find solace and positive developments in their behavior and mental health by visiting nature and wilderness areas. Visiting wild areas and camping, mountain climbing, rafting, fishing, and hiking enables a sense of adventure that requires policies that protect land from over-development. The wilderness is an area where the Earth and its community of life are untrammeled by man, where man himself is a visitor who does not remain [30]. Yellowstone National Park was established in 1872 to protect its geysers, waterfalls, and bison. In 1906, the Antiquities Act allowed wilderness, especially those containing cultural artifacts, to be declared National Monuments by Presidential Executive Order. The U.S. Forest Service was established in 1905, and its first protected roadless area, the Gila Wilderness in New Mexico, was established by Aldo Leopold in 1924. The Wilderness Society was founded in 1936 by Robert Marshall and others to influence public lands policies to protect wilderness as a resource. In the 1930–1960s, the Bureau of Reclamation was damming rivers throughout the West to develop irrigation for farmers and hydroelectricity. When they proposed damming the Yampa River flooding the Dinosaur National Monument, a bipartisan effort was begun to write a wilderness bill proscribing dams in protected wild areas. The Wilderness Society helped write The Wilderness Act, which went through 66 versions in 18 hearings over 8 years, eventually being passed by Congress in 1964. An Act of Congress was required, with a recommendation from the President or federal agencies to create a wilderness area. Fifty-four areas within national forests were designated “wilderness” if they previously had terms of wilderness, primitive, or canoe totaling 9.1 million acres. Currently, over 111, 209, 349 acres covering 4.5% of U.S. land are protected under the Wilderness Act. There are 840 Wilderness Areas with 36.7 million acres in national forests, 20.7 million acres in national wildlife refuges, and 10 million acres managed by the Bureau of Land Management. Over half is in Alaska at 57.7 million acres. About half of the 63 National Parks (44 million acres) are protected as wilderness. Most of the wilderness areas are in the west, but the Eastern Wilderness Act of 1975 created several Eastern wilderness areas. State parks created as wilderness included New York’s Catskills and Adirondack State Parks, the latter being enshrined in the State Constitution as forever ‘wild’. Maine also created the Baxter State Park, protecting Mt. Katahdin, and the Allagash Wilderness Waterway, protecting 92 miles of the Allagash River since 1966.

## 6. Conclusions

The revamped course has proved very successful in fostering student interactions, focusing on unique global environmental threats with the aim to critically evaluate the challenges and develop solutions, and present the results in systems maps, memos, and slide presentations. At the end of the course, instead of a final examination, they write a reflection on their experience. One of these conclusions is an example of a student memo: Memo 2.

**Memo 2:** *Personal Final Paper Reflections*.

This course was an enlightening experience into the world of environmental health, which encompassed many more aspects than I initially thought. The thought process behind a systems map and just how many different variables could affect any one topic was eye-opening and was something that I kept in mind throughout all the projects. The roles that we were assigned also showed me how many different perspectives could come into play when looking at any issue, and how those differing perspectives come together when it comes to finding a solution: What are the environmental and epidemiological effects? Is it economically possible? Is it politically possible? Is it possible from an engineering point of view? There were a lot of different aspects that had to be taken into consideration. I enjoyed contributing within my given role, with each role being harder or easier depending on the topic. Some roles had more of a stake in an issue than others, which was interesting to see. It was challenging to be an engineer, for example, if the solution to the problem was not a simple “just create something for this.” It was also challenging to be an epidemiologist when the issue does not revolve around a disease. I will say that if I had a personal favorite role, it would most likely be the epidemiologist. However, even if some roles had more of a stake than others, every role in each of the topics did have some sort of role to play. Working together as a group, each with different roles, and discussing solutions to issues depending on what we researched through the lens of each of our roles was an enriching experience. Regarding specific topics, I enjoyed learning about how air pollution is ranked globally with the Air Quality Index, and it was interesting to analyze how different countries/regions dealt with the issue. My group focused on Beijing, China, for that topic, and I thought it was interesting that the issue in Beijing was not solely human, but the issue was perpetuated by the topography of the region. There were economic variables, epidemiological effects, and lots of other related factors that were interesting to discuss. The discussion on planetary health was also quite interesting as I learned a lot about these beautiful places around the US that I did not even know existed, such as Bears Ears, and about issues surrounding the protection of these places. The effects of the policies of different presidents on these beautiful areas were interesting too, and my own research into the effects that these natural regions could have on mental health was great to learn. Health security was also interesting as I learned a lot more about diseases that I had heard about but did not know much about. For example, I had been aware of Zika when it was first discovered, as many others my age were, and yet I learned a lot more about the symptoms and treatments associated with it. The Guns assignment was also interesting as I had heard of organizations that are pro-guns, such as the NRA, and yet I did not have much knowledge about groups that were focused on lowering gun violence and the like, such as the Brady Campaign. This assignment helped increase my knowledge of the stakeholders at play. Regarding the Flint lead and water assignment, I remember being assigned to the Nitrates and Phosphates group, and I had no idea how this related to water or could cause an issue, and yet these elements were very important to consider when thinking about fertilizers, run-off, eutrophication, and algae blooms. It was an issue that I did not even know was an issue, so this specific assignment was very informative. Overall, the variety of different topics, the presentations of my classmates and the research and work put into my own, and the lectures from Professor Rom were all incredibly helpful in Environmental Health in a global context. I am now aware of these issues from a variety of different roles, and if I were to choose to pursue a career in any of these roles, whether that be an epidemiologist, an environmental scientist, or even an economist, I would now have a baseline for what that would entail. This course was a great insight into the issues of environmental health.

## Figures and Tables

**Figure 1 ijerph-20-06146-f001:**
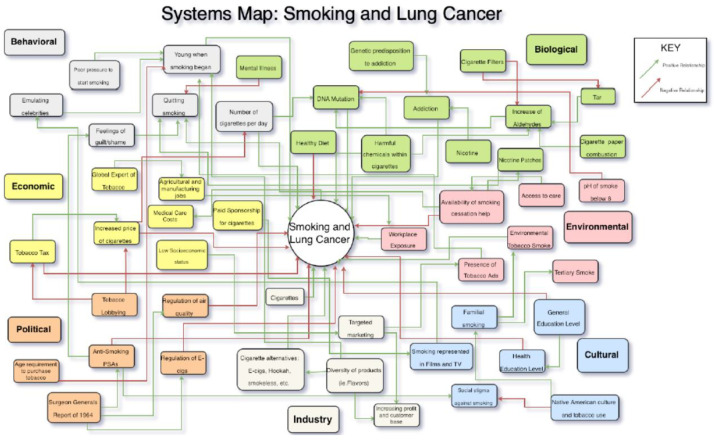
Figure systems map smoking and lung cancer. The Figure was rotated by me to take a full page to make it more legible.

## Data Availability

Data is available from Corresponding author.

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
