# Peer review of "A New Model of Learning: Environmental Health in a Global World"

_ijerph, 2023, doi:10.3390/ijerph20126146_

Round 1

Reviewer 1 Report

The manuscript presents an innovative approach to environmental health education, focusing on systems thinking and visual representation of complex factors. The authors emphasize the importance of understanding the interactions between political, social, economic, and other factors in addressing environmental health challenges. Through a series of modules, students create systems maps and develop strategies to improve health outcomes. The manuscript also discusses the use of assigned roles and avatars to encourage multiple perspectives and cooperation within teams. The results of implementing this revamped course are presented, highlighting the positive response from students and the high standard of performance maintained throughout the semester.

I thoroughly enjoyed reading this manuscript and found the approach presented to be highly commendable. The authors have successfully designed a course that not only imparts knowledge but also develops critical thinking skills and a deep understanding of environmental challenges. The incorporation of systems thinking and visual representation techniques is particularly praiseworthy, as it allows students to analyze the complexity of environmental health issues comprehensively.

One of the notable strengths of this manuscript is the inclusion of assigned roles and avatars, which foster a multidisciplinary approach to problem-solving. By assigning students different perspectives, such as anthropologist, economist, policy/government official, epidemiologist, biologist, and environmental scientist, the authors ensure that teams consider multiple viewpoints and engage in cooperative research. This approach undoubtedly enhances the quality of their systems maps and strategies.

The results presented in the manuscript demonstrate the effectiveness of the course in engaging students from various disciplines. The wide spectrum of students who participated, with dual majors in sociology, anthropology, nutrition, nursing, biology, social work, media, and history, reflects the course's interdisciplinary appeal and relevance. The fact that the course has been successfully taught to more than 180 students each year speaks volumes about its popularity and impact.

Furthermore, the selection of module topics, ranging from smoking and lung cancer to climate change and planetary health, showcases the breadth and significance of the environmental challenges addressed in the course. The authors have ensured that students focus on specific geographical regions and at-risk populations, enabling them to develop context-specific strategies for improving health outcomes.

In conclusion, this manuscript offers a compelling approach to environmental health education. The integration of systems thinking, visual representation, assigned roles, and multidisciplinary perspectives creates a stimulating learning environment that encourages critical thinking and problem-solving. The positive results achieved thus far, along with the high level of student engagement, further validate the effectiveness of this approach. I highly recommend this manuscript for publication as it contributes valuable insights to the field of environmental health education.

 Hovewer, Figure 1 should be rotated by 90 degrees and occupy the entire page to be more readable.

References need to be adapted to the requirements of the journal.

Author Response

The manuscript presents an innovative approach to environmental health education, focusing on systems thinking and visual representation of complex factors. The authors emphasize the importance of understanding the interactions between political, social, economic, and other factors in addressing environmental health challenges. Through a series of modules, students create systems maps and develop strategies to improve health outcomes. The manuscript also discusses the use of assigned roles and avatars to encourage multiple perspectives and cooperation within teams. The results of implementing this revamped course are presented, highlighting the positive response from students and the high standard of performance maintained throughout the semester.

The authors appreciate Reviewers' comments.

 Hovewer, Figure 1 should be rotated by 90 degrees and occupy the entire page to be more readable.

References need to be adapted to the requirements of the journal.

Figure rotated and references adapted to meet the requirements of the journal.

Reviewer 2 Report

Overall, very important and creative way to teach environmental health.  I would not describe it as a research study per say, but as a case study. You may want to clarify how teams are created and that the topics a based on the 7 content modules (i.e. when you discuss concerns about it on ln 71). How are the assignments graded? 

See below for specific comments:

Pg 3:

ln 90: Fig 1, some font is very small

ln 92: The Pb in drinking water is not all Pb related

ln 110: memo needs to be better attributed, it is a large portion of the paper

Pg 5: the discussion really belongs in the methods because it just describes course content. 

Author Response

Overall, very important and creative way to teach environmental health.  I would not describe it as a research study per say, but as a case study. You may want to clarify how teams are created and that the topics a based on the 7 content modules (i.e. when you discuss concerns about it on ln 71). How are the assignments graded? 

The teams are assigned at random for the first two modules, then re-assigned for the third and fourth modules, and finally re-assigned for the last three modules.  In special cases we take into consideration special requests for specific assignments.  There are 4 points each for the presentation, memo, and systems maps for the seven modules plus 16 points for the final reflection memo. ln 90: Fig 1, some font is very small We rotated this Figure to take an entire page.    ln 92: The Pb in drinking water is not all Pb related We discuss other sources of exposure-see references on Fuller, Needleman and Bridbord. ln 110: memo needs to be better attributed, it is a large portion of the paper Both memos are attributed as an example of students' work. Pg 5: the discussion really belongs in the methods because it just describes course content.  We considered this also but ended up preferring this discussion in the Discussion.

Round 2

Reviewer 2 Report

I respectfully disagree with your comments about the student groups related to Pb and water (nitrates, PFAS and fracking are not related to Pb) and also with your decision to have the basic content as the discussion. It would be chronological to have the course content come before the description of the student groups to provide context for their work. However, in the end, you are the authors.    

Author Response

The Reviewer lists the Abstract and we agree with its content.  A New Model of Learning:  Environemtntal Health in a Global World

There is an online supplement of a student slide presentation.